**Data Availability Statement:** All relevant data are within the paper and its Supporting Information files.

# Chest pain in pediatric patients in the emergency department- Presentation, risk factors and outcomes-A systematic review and meta-analysis

**Mohammed Alsabri**[1]*, **Alaa Ahmed Elshanbary**[2]*, **Anas Zakarya Nourelden**[3], **Ahmed Hashem Fathallah**[4], **Mohamed Sayed Zaazouee**[5]*, **Jorge Pincay**[6], **Zaid Nakadar**[6], **Muhammad Wasem**[7], **Lita Aeder**[8]

1 Department of Pediatrics/Emergency Pediatrics, Althawra Modern General Hospital, Sana'a, Yemen, 2 Faculty of Medicine, Alexandria University, Alexandria, Egypt, 3 Faculty of Medicine, Al-Azhar University, Cairo, Egypt, 4 Faculty of Medicine, Minia University, Minia, Egypt, 5 Faculty of Medicine, Al-Azhar University, Assiut, Egypt, 6 SUNY Downstate Health Sciences University, Brooklyn, NY, United States of America, 7 Pediatric Emergency Department NYC Health + Hospitals/Lincoln, Bronx, NY, United States of America, 8 Pediatric Department, Brookdale Hospital, Brooklyn, NY, United States of America

* alsabritop@yahoo.com (MA); Alaa.mohamed1603@alexmed.edu.eg (AAE); MohamedZaazouee.stu.6.44@azhar.edu.eg (MSZ)

## Abstract

### Objective

This study aimed to assess and determine the presentation, risk factors, and outcomes of pediatric patients who were admitted for cardiac-related chest pain.

### Background

Although chest pain is common in children, most cases are due to non-cardiac etiology. The risk of misdiagnosis and the pressure of potentially adverse outcomes can lead to unnecessary diagnostic testing and overall poorer patient experiences. Additionally, this can lead to a depletion of resources that could be better allocated towards patients who are truly suffering from cardiac-related pathology.

### Methods

This review was conducted per PRISMA guidelines. This systematic review used several databases including MEDLINE, Embase, Scopus, and Web of Science to obtain its articles for review.

### Results

A total of 6,520 articles were identified, and 11 articles were included in the study. 2.5% of our study population was found to have cardiac-related chest pain (prevalence = 0.025, 95% CI [0.013, 0.038]). The most commonly reported location of pain was retrosternal chest pain. 97.5% of the study population had a non-cardiac cause of chest pain, with musculoskeletal pain being identified as the most common cause (prevalence = 0.357, 95% CI

**Funding:** The authors received no specific funding for this work.

**Competing interests:** The authors have declared that no competing interests exist.

**Abbreviations:** CP, Chest Pain; NOS, Newcastle Ottawa Scale; ED, Emergency Department; ICU, Intensive Care Unit.

[0.202, 0.512]), followed by idiopathic (prevalence = 0.352, 95% CI [0.258, 0.446]) and then gastrointestinal causes (prevalence = 0.053, 95% CI [0.039, 0.067]).

## Conclusions

The overwhelming majority of pediatric chest pain cases stem from benign origins. This comprehensive analysis found musculoskeletal pain as the predominant culprit behind chest discomfort in children. Scrutinizing our study cohort revealed that retrosternal chest pain stands as the unequivocal epicenter of this affliction. Thorough evaluation of pediatric patients manifesting with chest pain is paramount for the delivery of unparalleled care, especially in the context of potential cardiac risks in the emergency department.

## Background

Chest pain stands as one of the most frequently encountered complaints in emergency departments worldwide. While its etiology can span a spectrum from benign to life-threatening, it is crucial to distinguish between these categories for effective patient management. Serious causes of chest pain encompass aortic dissection, pulmonary embolism, and pneumothorax, demanding immediate attention.

On the other hand, non-emergent cardiac factors contributing to chest pain encompass valvular, infectious, or inflammatory conditions. Interestingly, a Study by Chen L et al. aimed to investigate the causes of chest pain in Chinese children and found that causes of chest pain in 7,251 children who visited the hospital during a 15-year period; 53.0% of the cases were idiopathic, 29.1% were related to Musculoskeletal diseases, 9.1% were related to respiratory diseases, 8.0% were cardiac, 0.6% were related to gastrointestinal diseases, 0.16% were related to mental diseases, and 0.04% were related to other conditions [1]. Nevertheless, according to this study, idiopathic chest pain was the most common cause of chest pain [1]. Muscloskeletal chest pain, strains, psychiatric conditions (including panic attacks), and systemic inflammatory and rheumatologic disorders round out the spectrum of potential causes in children, requiring thorough evaluation [1]. According to Januzzi et al., the cost of chest pain totals over 5 billion USD, with over 7 million presentations in emergency departments annually. Cardiovascular causes may be in up to 20% of patients presenting with chest discomfort; only 5.5% of these patients have an acute life-threatening condition, whereas more than 50% of patients presenting with chest discomfort receive a diagnosis of noncardiac pain [2]. On one hand, proper evaluation of chest pain must be done to rule out any significant issues, such as a myocardial infarction. Pope et al. discussed the complications of missed interventions, as missed myocardial infarctions can lead to a doubled risk of mortality after discharge [3]. On the other hand, prolonged assessment of all patients who present to the emergency department with chest pain is burdensome to all facets of the healthcare system. According to McCullough et al., the valuation per quality-adjusted life year (QALY) is over USD100,000 and would still be considered highly cost-ineffective compared with many other strategies and treatments for cardiovascular patients [4]. The goal of care should be to have a criterion that allows for efficient evaluation without unnecessary expenditures and waste of resources.

Chest pain in children without a prior history of cardiac abnormalities or significant family histories is unlikely to be related to a cardiac cause. According to a research article published in the Journal of Pediatric Health Care, a study of 50 children presenting to cardiology with chest pain had the following breakdown: 38 children (76%) had musculoskeletal/

costochondral chest pain, 6 children (12%) had exercise-induced asthma, 4 children (8%) had chest pain resulting from gastrointestinal causes, and 2 children (4%) had chest pain resulting from psychogenic causes [5]. In an article written by Dr. Selbst in the Pediatric Clinics of North America journal, the approach to a child with chest pain should be taken seriously because the symptom of chest pain can cause great anguish to the child, disturbing their daily life, and can also create strain on the caretakers due to worry about a more malicious or dangerous pathology [6].

While chest pain in the pediatric population is even more rarely associated with a true cardiac etiology, some important information must be taken into consideration when reviewing past medical records or taking a history [7, 8]. The first would be a history of congenital heart disease. These conditions are often evaluated at birth and are noted for further management or observation in benign cases. Post-congenital causes of pediatric cardiac issues can include infections that can lead to cardiac inflammation, along with more systemic syndromes such as Rheumatic Fever and Kawasaki disease [6–8]. Genetic abnormalities and gene mutations can also lead to specific cardiac issues, such as dilated or hypertrophic cardiomyopathy and even a more systemic condition such as Marfan syndrome [6–8]. These conditions can all lead to chest pain being due to a cardiac etiology and can be effectively deduced through chart review, good history taking, and thorough physical examinations [7, 8]. The main objective of our study is to systematically review the literature that assesses the presentation, risk factors, and outcomes of pediatric patients who were admitted for cardiac chest pain.

## Methods

This review was conducted per PRISMA guidelines [9]. All steps were done per the Cochrane Handbook of Systematic Reviews and Meta-analysis of Interventions [10]. Our research question was developed following the key elements of the PICO framework: Participants, Interventions, Comparison, and Outcomes [11]. The protocol was registered in PROSPERO), under number CRD42023397158

### Inclusion and exclusion criteria

Both qualitative and quantitative studies were sought. Primary articles that focused on assessing the presentation, risk factors, and outcomes of pediatric (less than 21 years) patients who present to the ED with cardiac chest pain. In particular, articles that mentioned the incidence of cardiac-related chest pain were included in order to compare statistically to the general population of pediatric non-cardiac chest pain in the ED. Time limits were set from 2000 to the present and were included. Secondary or tertiary articles were excluded. Articles published before 2000 and non-English literature were excluded. Studies on the CP adult population (>21 years old), animals, or case series were also excluded. The list of excluded studies is included in the supplementary information. We also excluded studies performed only in urgent care, primary care, emergent medical services, or the ICU. There were no other restrictions regarding the type of ED, and studies performed in urban and community hospitals, public and private ED, academic and non-academic ED, trauma-center level I, II, and III will be included. Any patients aged 21 years or above who present to the ED with complaints of chest pain were excluded.

### Data sources and study selection

The literature search strategies were developed using medical subject headings (MeSH) and text words related to chest pain in pediatrics. The following databases were queried for identifying peer-reviewed literature: MEDLINE, EMBASE, SCOPUS, and Web of Science. To ensure

literature saturation, we scanned the reference lists of included studies and relevant reviews identified through the screening. Finally, we provided a bibliography of the included articles to the systematic review team. The most recent search was conducted in July 2023. The keywords used for the search are included in the attached supplement (S1 Appendix).

Screening was completed in two stages using the systematic review management program. Articles were screened for relevance based on the title and abstract and then evaluated for inclusion based on the full text. Two reviewers (M.S.Z and A.A.E) independently screened the titles and abstracts. The selection was focused only on peer-reviewed published studies. The reviewers read the full-text articles obtained and selected those that met all inclusion criteria. A third author (M.A) assisted in resolving any disagreements through consensus agreement. We used the input of two researchers at all stages of the analysis. A graphic of the screening and selection process can be seen in Fig 1.

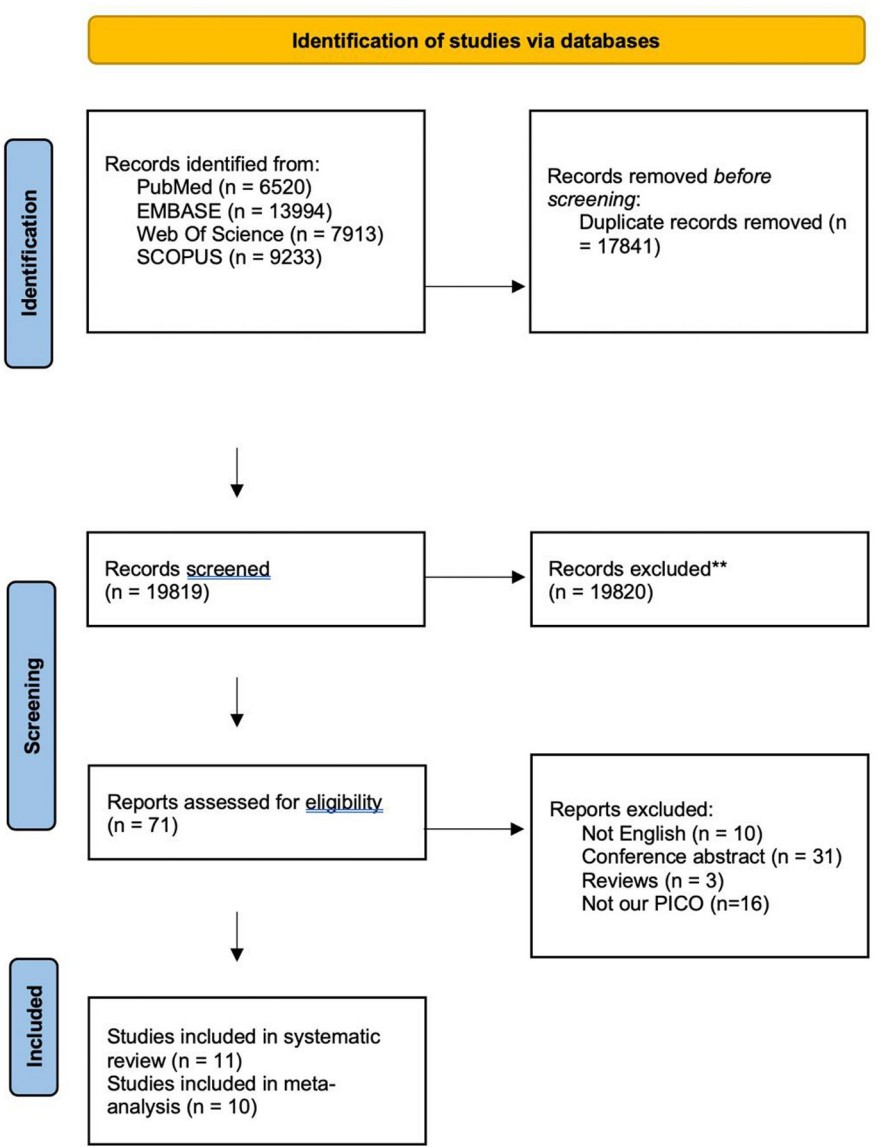

**Fig 1. Study selection process.** PRISMA flow diagram illustrating the process of study selection for the systematic review and meta-analysis.

## Quality assessment of studies

Two assessors independently rated the quality of the studies using the Cochrane Collaboration's tool. In addition, if discrepancies were presented, these were resolved through discussion and consensus between the analysts. The certainty of evidence was assessed using GRADE methodology [12].

## Data extraction

Data were collected on the study characteristics: design, setting, population, sample size, and main objective. Data on prognostic factors and outcomes were also extracted: demographics, clinical presentation, associated symptoms, and risk factors. Missing information on prognostic factors or outcome was requested from the authors.

## Data analysis

We extracted and analyzed the data regarding the prevalence of cardiac and non-cardiac emergency causes, which were pooled to calculate overall prevalence estimates with a 95% confidence interval (CI). The random effect model using the DerSimonian-Laird method was applied. We investigated the statistical heterogeneity between studies using the I2 statistics and chi-squared test, with $p < 0.1$ considered heterogeneous and I2 $\geq$ 50% suggestive of high heterogeneity. We conducted statistical analyses using Comprehensive Meta-Analysis Software (CMA).

# Results

## Results of the literature search

A total of 6520 records were identified through searching the four included databases, resulting in 19819 records after discarding the duplicates. Seventy-one records resulted from the title and abstract screening, and after the full-text screening, 11 records were finally included in our systematic review; of them, 10 studies were included in the meta-analysis [13–23]. The selection process is described in the PRISMA flow chart in Fig 1.

## Characteristics of the included studies

Our search retrieved 11 studies with 3,052,540 children suffering from chest pain. The reported mean age for studied patients ranged from 10.1 in Massin et al. to 13 years in Pissarra et al. and Lin et al. [18, 19, 23]. All our included studies were retrospective cohorts in design except for Massin et al., which was prospective, and Mohan et al., whose study was quasi-experimental [19, 20]. Only three of our studies reported the location or radiation of pain; Pissarra et al. revealed that the commonest chest pain location among their patients was retrosternal, and Massin et al. reported that two locations were left-sided and diffuse chest pain [19, 23]. However, Ocampo-Vázquez et al. reported that 58% of their patients showed no pain radiation [22]. Throughout the studies, ECG, chest radiography, and Echocardiography were the most demanded investigations. Characteristics and a summary of our included research are displayed in Tables 1 and 2, respectively.

## Risk of bias assessment

Most of our included studies demonstrated good quality on NOS assessment. However, most of them did not have an adequate follow-up period. S1 Table shows the detailed assessment of

**Table 1. Characteristics of the included studies.** Summary of studies on myocarditis and chest pain presentations, detailing study design, patient demographics, inclusion criteria, and observed symptoms. Beside the study ID the reference number of the study can be found.

| Study ID | Study Design | Study arm, n | Male, n (%) | Age, M (sd) | Site | Inclusion criteria | Medical history | Family History disease | Location or irradiation of the pain | Associated symptoms |
|---|---|---|---|---|---|---|---|---|---|---|
| **Babbitt 2022** [13] | Retrospective cohort study | 127 | NR | NR | USA | 1. Patients discharged from the PICU, with a discharge diagnosis of myocarditis or myopericarditis | NR | NR | NR | 1. Dyspnea, n = 23 |
| | | | | | | 2. The diagnosis of myocarditis was based on the clinical picture of acute deterioration of cardiac function requiring inotropic support, and endomyocardial biopsy was not required for the diagnosis of this study | | | | 2. Fever, n = 28 |
| | | | | | | 3. Myopericarditis was diagnosed based on chest pain during hospitalization, elevated troponin, and exclusion of other potential causes, such as coronary artery abnormalities. | | | | 3. Abdominal pain, n = 11 |
| | | | | | | | | | | 4. Vomiting, n = 15 |
| | | | | | | | | | | 5. URI symptoms, n = 24 |
| **Brancato 2021** [14] | Retrospective cohort study | 99 | 61 (62.9%) | 12.5 | Italy | 1. Patients aged 0 to 19 years admitted for chest pain to the pediatric ED. | Only 6 (23.1%) of 26 patients who presented a medical history and suggestive targets of cardiac pathology received a diagnosis of cardiac pathology | Personal history of cardiac disease in 3 patients (3%) (2 with a small patent ductus arteriosus and 1 with previous closure of an atrial septal defect), tachycardia in 4 patients (4%), and palpitations in 4 patients (4%). | NR | 1. Syncope in 3 patients (3%) |
| | | | | | | 2. Patients were included in the study if they had known minor congenital heart disease including left to right shunts and resolved lesions, or if they had no diagnosed lesion at the time of presentation. | | | | 2. Association with fever and asthenia in 4 patients (4%) |
| | | | | | | | | | | 3. Chest pain during physical activity in 8 patients (8.1%) |
| **Drossner 2011** [15] | Retrospective cohort study | 4436 | 2237 (53%) | 11 (3.66) | USA | 1. Age less than 19 years with a chief complaint of chest pain. | Final review yielded 3% (148/4436) of patients excluded secondary to a documented history of congenital/acquired cardiac disease. | NR | NR | NR |
| **Gesuete 2020** [16] | Retrospective cohort study | 737 | 430 (59%) | 11 (1.16) | Italy | 1. Pediatric patients who presented with chest pain and were evaluated in a pediatric ED | History of cardiac disease (n = 24) | NR | NR | Cardiac-related Chest pain N = 8; |
| | | | | | | | | | | 1. Fever 3 (37.5%) |
| | | | | | | | | | | 2. Palpitations 0 (0%) |
| | | | | | | | | | | 3. Exertional dyspnea 0 (0%) |
| | | | | | | | | | | 4. Headache 0 (0%) |
| | | | | | | | | | | 5. Gastrointestinal symptoms 0 (0%) |
| | | | | | | | | | | Non-cardiac chest Pain N = 729; |
| | | | | | | | | | | 1.Fever 130 (17.8%) |
| | | | | | | | | | | 2. Palpitations 31 (4.2%) |
| | | | | | | | | | | 3. Exertional dyspnea 14 (1.9%) |
| | | | | | | | | | | 4. Headache 15 (2%) |
| | | | | | | | | | | 5. Gastrointestinal symptom 55 (7.5%) |

(Continued)

**Table 1.** (Continued)

| Study ID | Study Design | Study arm, n | Male, n (%) | Age, M (sd) | Site | Inclusion criteria | Medical history | Family History disease | Location or irradiation of the pain | Associated symptoms |
|---|---|---|---|---|---|---|---|---|---|---|
| Hambrook 2010 [17] | Retrospective cohort study | 3,043,873 | 1,361,611 (45%) | 0–5 years 35019012, 6–11 years 74324124, 12–18 years 95044264 | USA | 1. Their age was less than 19 years during an ED visit for a chief complaint of CP. | NR | NR | NR | NR |
| Lin 2008 [18] | Retrospective cohort study | 103 | 64 (62.1%) | 13 (2.17) | Taiwan | 1. Children under the age of 18 years who presented to the ED with the chief complaint of chest pain | Seven (6.9%) patients reported a history of asthma. Two patients had mental retardation. | Sudden death had been experienced in one family only. Two patients reported some psychogenic problems in their family members. | NR | 1. Fever or respiratory symptoms (cough, dyspnea) in 34 2. Gastrointestinal symptoms (epigastric pain, nausea, vomiting) in eight, dizziness in five, and palpitation in three. |
| Massin 2004 [19] | Prospective cohort study | GROUP A, n = 168 | 90 (53.57%) | 10.1 (3.4) | Belgium | 1. Children younger than 16 years of age who presented to the emergency department with a primary complaint of chest pain | One child in the cardiac category had a history of beta-thalassemia major and secondary hemochromatosis.14 One patient in the digestive category and 5 in the CW category had asthma; 1 case was diagnosed a few months after the presentation to the hospital. | The family history was usually uneventful: the father of a child in the CW category had undergone an aortic valvuloplasty and the father of a child in the psychogenic group underwent a cordectomy. | 1. Left-sided = 60 2. Right-sided = 18 3. Diffuse = 48 4. Sternal = 32 | 1. Anxiety 3 2. Fever 13 3. Cough 16 4. Dyspnea 6 5. Palpitations 19 6. Headache 1 7. Depression 1 8. Stressful event 12 9. Hyperventilation 1 10. Pain reproduced by palpation 23 11. Previous trauma 8 12. Pallor 4 13. Pathologic heart auscultation 18 14. Exercise-related pain 6 15. Meal-related pain 3 16. Vomiting 1 17. Hematemesis 1 |
| | | GROUP B, n = 69 | 43 (62.31%) | 10.6 (2.9) | | | The personal history was uneventful for the children in the CW category, except for one who had asthma and another one with a history of supraventricular tachycardia during infancy. One child in the cardiac group had a history of supraventricular tachycardia at birth without demonstrated recurrence beyond the neonatal period | The family history of all the patients was uneventful. | 1. Left-sided = 10 2. Right-sided = 0 3. Diffuse = 55 4. Sternal = 4 | |
| Mohan 2018 [20] | A quasi-experimental design | Pre-Pathway, n = 670 | 294 (43.6%) | 12.4 (1.07) | USA | 1. Children aged 3 to 18 years old presenting with chest pain to the ED | Patients were flagged for exclusion if they had a known alternative etiology for chest pain, such as sickle cell disease, gastroesophageal reflux (GER), or asthma, before the ED encounter. Patients who also had a known cardiac diagnosis or diagnosis that is known to have a high likelihood of presenting with chest pain (i.e. Marfan's Syndrome) were also excluded | NR | NR | 1. Palpitations 22 2. Arrhythmia 2 |
| | | Post-Pathway, n = 1012 | 428 (42.3%) | 12.1 (2.65) | | | | | | |

(*Continued*)

**Table 1.** (Continued)

| Study ID | Study Design | Study arm, n | Male, n (%) | Age, M (sd) | Site | Inclusion criteria | Medical history | Family History disease | Location or irradiation of the pain | Associated symptoms |
|---|---|---|---|---|---|---|---|---|---|---|
| **Neff 2012** [21] | Retrospective cohort study | 400 | 201 (50.1%) | 10.7 | USA | 1. Patients 18 years or younger presenting to a pediatric ED with a chief complaint of chest pain | There were a significant number of patients with preexisting comorbid conditions, the most common of which were asthma and congenital cardiac anomalies. | NR | NR | NR |
| **Ocampo-Vázquez 2019** [22] | Retrospective cohort study | 48 | 24 (50%) | 10.9 (2.2) | Mexico | 1. Patients oscillated between 0 and 17 years without a clear predominance regarding the sex of the patients. 2. All of them went to the pediatric ER service to get a consult regarding precordial pain. | All patients with previous cardiac pathologies and patients with chronic musculoskeletal problems under treatment were excluded from the study | A total of four cases with a positive family history were found, these four cases had a family history of hypercholesterolemia, and in one case, a non-specified arrhythmia was found, in one other case, apart from the family background of hypercholesterolemia, the father also presented a family history of mitral insufficiency | 1. Without irradiation n = 28 (58.3%) 2. Retrosternal n = 8 (16.6%) 3. Left arm n = 1 (2%) 4. Left hypochondrium n = 3 (6.2%) 5. Left hemithorax n = 2 (4.1%) 6. Neck n = 1 (2%) 7. Superior limbs n = 1 (2%) 8. Right arm n = 1 (2%) 9. Right hypochondrium n = 1 (2%) 10. Right hemithorax n = 2 (4.1%) | 1. None n = 19 (39.5%) 2. Dyspnea n = 17 (35.4%) 3. Dysphagia n = 6 (12.5%) 4. Syncope n = 5 (10.4%) 5. Anxiety n = 1 (2%) 6. Headache n = 1 (2%) 7. Diaphoresis n = 1 (2%) 8. Palpitations n = 1 (2%) 9. Paresthesia n = 1 (2%) 10. Heartburn n = 1 (2%) 11. Cough n = 1 (2%) 12. Vomit n = 1 (2%) |
| **Pissarra 2022** [23] | Retrospective cohort study | 798 | 370 (46.4%) | 13 (2.67) | Portugal | 1. All admissions in pediatric ED with a chief complaint of chest pain | 32.5% psychiatric (including a history of anxiety, depression, and attention deficit hyperactivity disorder), 28.3% respiratory (including a history of wheezing or asthma), and 18.9% cardiovascular (including a history of congenital heart defects or arrhythmia). | NR | 1. 33.7% mentioned it was a retrosternal pain 2. 23.6% in the left hemithorax 3. 16.7% an anterior pain 4. 7.1% in the right hemithorax 5. 3.2% a posterior pain 6. 1.1% a subcostal pain. | 65.3% of the patients mentioned at least 1 additional symptom, in descent order of frequency: dyspnea, cough, palpitations, gastrointestinal symptoms (including abdominal pain, heartburn, nausea, and vomiting), dizziness or hypotonia, fever, asthenia or tiredness, and syncope. |

**Table 2. Clinical summary of the included studies.** Overview of studies examining cardiac and non-cardiac chest pain presentations, including methodologies, findings, and conclusions. Beside the study ID the reference number of the study can be found.

| Study ID | ED-work-up (Resources) | ECG findings | Echocardiographic Findings | ED-initial diagnosis | Final/discharge diagnosis | Cases of cardiac chest pain | Cases of non-cardiac chest pain | Others | Conclusion |
|---|---|---|---|---|---|---|---|---|---|
| **Babbitt 2022 [13]** | 1. ECG 2. Echocardiogram 3. Chest radiographs (CXRs) | 58% had ST changes on initial ECG in cardiac chest pain patients | NR | NR | 1. Cardiac chest pain, n = 58 2. Non- Cardiac chest pain, n = 69 | 1. Myopericarditis, n = 36 2. Myocarditis, n = 22 | NR | NR | "Myopericarditis is a relatively common cause of chest pain for patients admitted to the pediatric intensive care unit, presents differently than true myocarditis, and carries a good prognosis" |
| **Brancato 2021 [14]** | 1. ECG 2. Troponin 3. Echocardiographic | The ECG pattern was pathological in 23 (23.2%) of 99 patients. In particular, 14 patients showed ST changes or repolarization abnormalities, and 9 patients presented dysrhythmia. Of the 9 patients with dysrhythmia, 3 showed runs of paroxysmal supraventricular tachycardia (TPSV), 1 run of ventricular tachycardia, and 1 showed Wolff-Parkinson-White, and 4 showed premature ventricular contractions. No pathological Q waves were detected in any patient | Echocardiographic examination was performed in all 9 cases with pathological ECG and/or TN of >0.03 ng/ mL. | 1. Yellow code, n = 66 (68%) 2. Green code, n = 32 (32%). Signs suggestive of cardiac pain (red flags) were detected in 26 patients (26.3%). Elements suggestive of respiratory pathology were found in 19 patients (19.2%). Medical history and physical examination suggestive of gastrointestinal pathology was found in 4 patients (4%). Two patients (2%) presented elements suggestive of psychogenic pathology. Twenty-eight patients (28.3%) did not refer to a specific etiology. | 1. Idiopathic chest pain in 45 patients (45.4%) 2. Musculoskeletal in 20 patients (20.2%) 3. Respiratory pathology in 19 patients (19.3%) 4. Cardiovascular pathology in 6 patients (6.1%) 5. Psychogenic pathology in 5 patients (5%) 6. Gastrointestinal pathology in 4 patients (4.0%) | NR | NR | A total of 8 patients were then excluded: 2 for a history of cardiac surgery within the last year, 5 for previous Kawasaki disease, and 1 because affected by aortic stenosis and waiting for surgery. | "Even with the low cost and the relative easiness for the plasma level determination, TN should be measured only in children with a familiar history suggestive of cardiovascular disease and/or clinical symptoms and/or ECG alterations" |
| **Drossner 2011 [15]** | Cardiac-related chest pain n = 24; 1. Chest radiograph 12 (50%) 2. Electrocardiogram 22 (92%) 3. Echocardiogram 14 (58%) 4. Laboratories 19 (79%) Noncardiac chest pain n = 4264; 1. Chest radiograph 2282 (53%), 2. Electrocardiogram 1171 (27%) 3. Echocardiogram 51 (1%) 4. Laboratories 1118 (26%) | NR | NR | Seventy-three patients were initially identified as having cardiac-related chest pain. | 1. Noncardiac chest pain was 56% of musculoskeletal disorders such as precordial pain, Tietze syndrome, and chest pain not otherwise specified (NOS). 2. Twelve percent were related to wheezing, asthma, and cough. 3. Eight percent of patients had infectious causes such as upper respiratory illness, pneumonia, fever, and pharyngitis. 4. Six percent of gastrointestinal-related problems such as esophagitis, gastritis, or abdominal pain. 5. Finally, 4% of patients with noncardiac chest pain were related to sickle cell anemia. 6. The major breakdowns of cardiac diseases by category were dysrhythmia, 37% (9/24); pericardial disease, 29% (7/24); myocarditis, 17% (4/24); acute myocardial infarction, 13% (3/24); and pulmonary embolism, 4% (1/24). | 1. Pericarditis, n = 6 2. Myocarditis, n = 4 3. Myocardial infarction, n = 3 4. Supraventricular tachycardia, n = 7 5. Long QT, n = 1 6. Ventricular tachycardia, n = 1 7. Pulmonary embolism, n = 1 8. Pneumopericardium, n = 1 | NR | NR | "Cardiac-related chest pain in pediatric patients is rare but potentially serious. Arrhythmia was the most common cardiac-related etiology among this cohort. Those with myocarditis and myocardial infarction were the most acutely ill. An electrocardiogram in addition to history and physical examination was most useful in detecting relatively uncommon but significant cardiac-related chest pain. Using a thorough physical examination and potentially an electrocardiogram evaluation by a pediatric emergency care physician has an excellent rate of detection of cardiac-related causes" |
| **Gesuete 2020 [16]** | Cardiac-related Chest pain N = 8; 1. Chest radiograph 4 (50%) 2. Electrocardiogram 8 (100%) 3. Echocardiogram 8 (100%) 4. Blood tests 6 (75%) Noncardiac-related Chest pain N = 729; 1. Chest radiograph 117 (16%) 2. Electrocardiogram 145 (20%) 3. Echocardiogram 59 (8%) 4. Blood tests 56 (7%) | NR | NR | NR | 1. Cardiac-related chest pain n = 8 2. Non- Cardiac chest pain n = 729; Musculoskeletal disorders n = 510 Respiratory disorders n = 88 Psychological disorders n = 73 gastrointestinal disorders n = 58 | 1. Pericardial disease n = 7 (87%) 2. Dysrhythmia n = 1 (13%) | NR | Patients with a previously documented history of cardiac disease were excluded. Prior cardiac disease was defined as a known history of congenital/ acquired heart disease, dysrhythmia, or cardiac surgery. | "Chest pain is mainly due to benign causes and is a recurrent symptom in a high percentage of patients, associated with re-admission and school absenteeism" |

(*Continued*)

**Table 2.** (Continued)

| Study ID | ED-work up (Resources) | ECG findings | Echocardiographic Findings | ED-initial diagnosis | Final/discharge diagnosis | Cases of cardiac chest pain | Cases of non-cardiac chest pain | Others | Conclusion |
|---|---|---|---|---|---|---|---|---|---|
| **Hambrook 2010** [17] | 1. CBC (23%) 2. CXR (54%) 3. EKG (35%) | NR | NR | NR | 1. Hematologic (0.3%) 2. Psychiatric (2.2%) 3. Cardiovascular (2.8%) 4. Gastrointestinal (6.4%) 5. Other (8.2%) 6. Respiratory (9.4%) 7. Trauma/Muskuloskeltal (12.8%) 8. Infectious (21.1%) 9. Chest Pain NOS (36.8%) | NR | NR | NR | "Disparities exist in the ED care of pediatric patients with CP. Identification of such variations is important and provides an opportunity for targeted interventions that ensure delivery of high-quality, cost-effective health care for children" |
| **Lin 2008** [18] | 1. Chest radiography 101 2. Electrocardiography 87 3. Echocardiography 15 4. Panendoscopy 6 5. Complete blood count 64 6. Creatine kinase MB 51 | Eighty-seven (84.5%) patients had ECG study, and four (4.6%) of them showed abnormalities, including first-degree atrioventricular (AV) block, Morbitz type 1 second-degree AV block, premature ventricular contraction, and Wolff-Parkinson-White syndrome. | Echocardiograms were performed in 15 (14.6%) patients. No major anomaly was found; only six showed mitral valve prolapse. | NR | 1. Idiopathic 61 (59.2%) 2. Respiratory 25 (24.3%) 3. Musculoskeletal 7 (6.7%) 4. Gastrointestinal 6 (5.8%) 5. Cardiac 2 (2.0%) 6. Miscellaneous 2 (2.0%) | 1. Arrhythmia 2 | • Respiratory causes 1. Bronchitis 11 2. Pneumonia 5 3. Asthma 1 4. Pneumothorax 3 5. Hyperventilation 5 • Gastrointestinal causes 1. Gastritis 3 2. Gastroesophageal reflux 3 | NR | "The most common cause of chest pain prompting a child to visit the ED is idiopathic chest pain. The careful physical examination can reveal important clues and save many unnecessary examinations" |
| **Massin 2004** [19] | 1. ECG 2. Echocardiogram 3. Chest radiography 4. 24-hour Holter recording. | ECG (positive) = 83 / The ECG was relevant to the diagnosis in the 2 cases with arrhythmia on auscultation: one child had intermittent atrial ectopic tachycardia and a second had severe supraventricular extrasystole. | Positive echocardiography 0 / Positive echocardiography 0 | NR | • Group A 1. Chest-wall pain (CW) was the most common diagnosis (64%). 2. Other causes included pulmonary (13%) 3. Psychological (9%) 4. Cardiac (5%) 5. Traumatic (5%) 6. Gastrointestinal problems (3%) 7. And 1 case of herpes zoster. • Group B 1. CW pain was also the most common diagnosis (89%). 2. Supraventricular tachyarrhythmia and exercise-induced asthma were demonstrated in 5 (7%) and 3 (4%) patients | 1. Cardiac problems were identified in only 9 patients: 3 patients had supraventricular reentry tachycardia, 2 had mitral valve prolapse, and 4 had sick sinus syndrome, myocarditis with extrasystole, Still disease with pericarditis, and cardiac hemochromatosis with left heart failure respectively. / NR | NR | NR | "The most important tools in assessing a child with acute chest pain in an emergency department are thorough history and physical examination. Assessment of recurrent chest pain is more difficult; arrhythmia and allergic and exercise-induced asthma may be underestimated when investigations are not performed" |
| **Mohan 2018** [20] | 1. Echocardiogram 2. ECG 3. CXR 4. Troponin and CK | NR | NR | NR | 1. A small minority of patients (11 patients) over the entire period was found to have a cardiac etiology for their chest pain 2. Unspecified Chest Pain (328) 3. Other Chest Pain (295) 4. Tietze's Disease (190) 5. Unspecified Viral Infection (40) 6. Esophageal reflux (46) 7. Asthma Unspecified with acute exacerbation (26) 8. Acute Upper respiratory infection of unspecified site (33) | 1. Four had myocarditis/ myopericarditis 2. Two had pericarditis 3. Two had supraventricular tachycardia 4. Three presented with premature ventricular contractions (one was after blunt chest trauma from a motor vehicle accident). 5. Five of the eleven patients with cardiac disease presented post-pathway implementation and were all identified by the pathway. | NR | NR | "Overall, there was a low incidence of cardiac disease, and to our knowledge, no undiagnosed cardiac or life-threatening diagnoses. We believe this has led to general acceptance of the pathway in the ED and therefore there is an opportunity to further improve upon the pathway. As is the case for most pediatric conditions, efforts at improving quality of care will require ongoing measurement and modifications to our existing practice" |
| **Neff 2012** [21] | 1. 63.5% (n = 254) of the children underwent a CXR while in the ED as part of their evaluation. | NR | NR | NR | 1. Total Major Significance 19 (7.4%) 2. Total Moderate Significance 2 (0.8%) 3. Total Major/ Moderate 21 (8.26%) 4. Total Minor Significance 11 (4.3%) 5. Total Any Significance 32 (12.56%) 6. Total No Significance 222 (87.4%) | NR | NR | NR | "This pilot study demonstrates the potential for a decision rule to eliminate both cost and radiation exposure by using defined criteria to determine the need for a CXR in pediatric ED patients. We identified 8 simple criteria that would have identified all children who benefited from a CXR in this study. The next phase of this study will prospectively evaluate the utility of each of the criteria as part of a draft decision rule" |

(Continued)

**Table 2.** (Continued)

| Study ID | ED-work up (Resources) | ECG findings | Echocardiographic Findings | ED-initial diagnosis | Final/discharge diagnosis | Cases of cardiac chest pain | Cases of non-cardiac chest pain | Others | Conclusion |
|---|---|---|---|---|---|---|---|---|---|
| **Ocampo-Vázquez 2019** [22] | 1. In 30 of the 48 patients, an electrocardiogram was performed 2. Of 48 patients, 32 had a thorax X-ray performed | In 30 of the 48 patients, an electrocardiogram was performed, of those 30 only 8 received abnormal results. All these studies were reviewed by a pediatric cardiologist who diagnosed five right branch blockings, one first-degree atrioventricular blocking with the left branch blocking, one right branch blockings, one sinus arrhythmia, and one sinus bradycardia. A total of three echocardiograms were performed in ER service, two were normal, and one presented pulmonary pressure of 40 mmHg. | NR | NR | 1. Cardiac causes 1 (2%) 2. Musculoskeletal 20 (41.5%) 3. Respiratory 0 (0) 4. GastroIntestinal 14 (29%) 5. Psychological 13 (26.9%) | 1. Pulmonary arterial hypertension 1 (2%) | • Gastrointestinal causes 1. Gastroesophageal reflux disease n = 14 (29%) • Psychological 1. Anxiety crisis 7 (14.5%) 2. Idiopathic 6 (12.4%) | NR | "There was only one case associated with the presence of cardiac precordial pain pathology regarding pulmonary hypertension; this signified an incidence of 2% similar to what has been previously published in other articles" |
| **Pissarra 2022** [23] | 1. Electrocardiogram (ECG) was performed in 62.4% 2. Chest radiographs were ordered in 52.6% 3. Blood analysis was performed on 13% 4. Cardiac biomarkers were tested in 8.9% | All were evaluated by pediatric cardiology and performed ECG, with altered results in 77.7% | NR | NR | 1. Musculoskeletal: 33% 2. Idiopathic: 24.4% 3. Psychogenic: 21.6% 4. Pulmonary chest pain occurred in 12.8% 5. Gastrointestinal 5.4% 6. Traumatic1.5% 7. Cardiac chest pain was reported in 1.1% 8. Miscellaneous 0.2% | 1. Cardiac chest pain was reported in 1.1% of cases: 2 cases of arrhythmia, 2 of myocarditis, 3 of myopericarditis, and 2 of pericarditis. | 1. Pulmonary chest pain occurred in 12.8% of the episodes, including acute asthma or wheezing, atypical and bacterial pneumonia, pneumothorax, and 1 case of plastic bronchitis. | NR | "Opposing to the low priority level in triage, benign diagnosis found, and low hospital admissions, there was a high percentage of complementary diagnostic tests performed with few altered results. In psychogenic chest pain, there was a low post-discharge referral. The authors highlight the importance of clinical algorithms to reduce unnecessary tests performed and readmissions and improve orientation and follow-up, particularly in psychogenic etiology" |

each study and their score in the individual items of the three major domains of NOS (selection, comparability, and outcome).

## Quality of studies

The quality of the studies can be viewed in S1 Table. The quality of each study was assessed according to the Newcastle Ottawa Scale. The scores obtained during the analysis ranged from 6 to 8, categorizing all our studies as intermediate to high quality. The overall average NOS score for our studies was 6.7. Given full consideration of our studies, we would categorize the overall quality as intermediate.

## Cardiac versus non-cardiac chest pain causes and presentation

Although the cardiac causes of chest pain in our studied population were less common (2.5%), they should be considered in the evaluation of chest pain in order to be excluded. Thus, ECGs and echocardiograms were commonly obtained in patients presenting with chest pain. Pissarra et al. reported that obtained ECG results were abnormal in 77.7% of their patients with cardiac etiologies of chest pain, and Lin et al. reported that 84.5% of the patients had ECG abnormalities [18, 23]. Additionally, Brancato et al. reported that 23.2% of their patients' ECGs were abnormal [14]. Cardiac causes of chest pain included myocarditis, pericarditis, myopericarditis, myocardial infarction, pulmonary embolism, pneumopericardium, and rhythm disturbances (supraventricular tachycardia, long QT syndrome, premature ventricular contraction). Reporting of previous causes varied among studies: Drossner et al. revealed that six patients had pericarditis, four had myocarditis, three had myocardial infarction, and seven had supraventricular tachycardia [15]; Gesuete et al. showed that 87% of cardiac causes were due to pericardial diseases and 13% were due to dysrhythmia [16]; Massin et al. reported that cardiac problems were identified in only 9 out of 130 patients: three patients had supraventricular reentry tachycardia, two had mitral valve prolapse, and four had sick sinus syndrome, myocarditis with extrasystole, Still disease with pericarditis and cardiac hemochromatosis with left heart failure, respectively [19].

On the other hand, non-cardiac chest pain causes were more commonly identified than cardiac causes (97.5% vs 2.5%). The non-cardiac causes reported in our included studies were as follows: respiratory, gastrointestinal, psychogenic, musculoskeletal, and idiopathic. Of these causes, Lin et al. reported that respiratory problems in their patients were bronchitis in 11 patients, pneumonia in five, asthma in one, pneumothorax in three, and hyperventilation in five, while gastrointestinal causes were due to gastritis in three patients and gastroesophageal reflux in three patients also [18]. Ocampo-Vázquez et al. reported that 29% of patients suffered from gastroesophageal reflux disease, while 14.5% of patients had anxiety crises and 12.4% of patients had idiopathic psychological causes for non-cardiac chest pain [22].

## Meta-analysis findings

The prevalence of chest pain causes among the studied population visiting the emergency department was ranked as follows, from most to least common: non-cardiac, musculoskeletal, idiopathic, respiratory, psychogenic, gastrointestinal, and cardiac. Presented are a few specific findings of our analysis for each of these outcomes:

## Cardiac causes

Ten studies reported cardiac causes of chest pain among our studied population [15–20, 22, 23]. The pooled prevalence was (0.025, 95% CI [0.013, 0.038]). The pooled studies were

## Prevalence of cardiac chest pain

| Study name | Point estimate | Standard error | Variance | Lower limit | Upper limit | Z-Value | p-Value |
|---|---|---|---|---|---|---|---|
| Babbitt 2022 | 0.457 | 0.044 | 0.002 | 0.370 | 0.544 | 10.339 | 0.000 |
| Brancato 2021 | 0.061 | 0.024 | 0.001 | 0.014 | 0.108 | 2.536 | 0.011 |
| Drossner 2011 | 0.005 | 0.001 | 0.000 | 0.003 | 0.008 | 4.908 | 0.000 |
| Gesuete 2020 | 0.011 | 0.004 | 0.000 | 0.003 | 0.018 | 2.850 | 0.004 |
| Hambrook 2010 | 0.028 | 0.000 | 0.000 | 0.028 | 0.028 | 296.114 | 0.000 |
| Lin 2008 | 0.020 | 0.014 | 0.000 | -0.007 | 0.047 | 1.450 | 0.147 |
| Massin 2004 | 0.034 | 0.012 | 0.000 | 0.011 | 0.057 | 2.875 | 0.004 |
| Mohan 2018 | 0.007 | 0.002 | 0.000 | 0.003 | 0.010 | 3.326 | 0.001 |
| Ocampo-Vázquez 2019 | 0.020 | 0.020 | 0.000 | -0.020 | 0.060 | 0.990 | 0.322 |
| Pissarra 2022 | 0.011 | 0.004 | 0.000 | 0.004 | 0.018 | 2.979 | 0.003 |
| | 0.025 | 0.006 | 0.000 | 0.013 | 0.038 | 4.095 | 0.000 |

Meta Analysis

**Fig 2. Prevalence of cardiac chest pain.** Forest plot illustrating the prevalence of cardiac chest pain across different studies, showing point estimates with 95% confidence intervals and associated statistical parameters. Test for overall effect; p-value = 0.00004, Z-value = 4.0946. Heterogeneity; p-value > 0.001, $I^2$ = 98.7%.

heterogeneous with I2 and p-value = (98.7% and > 0.001, respectively). The forest plot for cardiac chest pain outcome is illustrated in Fig 2.

## Non-cardiac causes

Non-Cardiac chest pain was reported in 10 studies with pooled prevalence = (0.975, 95% CI [0.962, 0.987]) [13–20, 22, 23]. I2 and p values were (98.7%, and > 0.001, respectively) showing that pooled studies were heterogeneous. The forest plot for non-cardiac chest pain outcome is shown in Fig 3.

## Respiratory causes

The overall prevalence of respiratory chest pain outcome pooled from eight studies was (0.13, 95% CI [0.092, 0.168]) [14–20, 22, 23]. The pooled studies were heterogeneous (I2 = 98.6%, p > 0.001). The forest plot for this outcome is illustrated in Fig 4.

## Gastrointestinal causes

Nine studies reported gastrointestinal chest pain outcome in the studied population [14–20, 22, 23]. The pooled prevalence was (0.053, 95% CI [0.039, 0.067]). The pooled studies were heterogeneous with I2 and p-value = (98.4% and > 0.001, respectively). The forest plot for gastrointestinal chest pain outcome is illustrated in Fig 5.

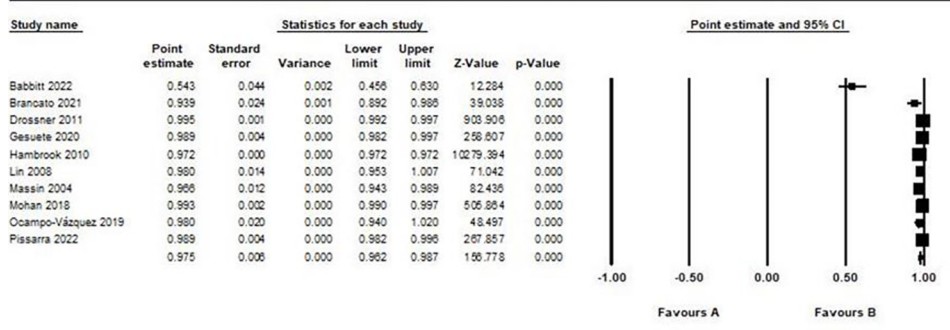

## Prevalence of Non-cardiac chest pain

| Study name | Point estimate | Standard error | Variance | Lower limit | Upper limit | Z-Value | p-Value |
|---|---|---|---|---|---|---|---|
| Babbitt 2022 | 0.543 | 0.044 | 0.002 | 0.456 | 0.630 | 12.284 | 0.000 |
| Brancato 2021 | 0.939 | 0.024 | 0.001 | 0.892 | 0.986 | 39.038 | 0.000 |
| Drossner 2011 | 0.995 | 0.001 | 0.000 | 0.992 | 0.997 | 903.906 | 0.000 |
| Gesuete 2020 | 0.989 | 0.004 | 0.000 | 0.982 | 0.997 | 258.607 | 0.000 |
| Hambrook 2010 | 0.972 | 0.000 | 0.000 | 0.972 | 0.972 | 10279.394 | 0.000 |
| Lin 2008 | 0.980 | 0.014 | 0.000 | 0.953 | 1.007 | 71.042 | 0.000 |
| Massin 2004 | 0.966 | 0.012 | 0.000 | 0.943 | 0.989 | 82.436 | 0.000 |
| Mohan 2018 | 0.993 | 0.002 | 0.000 | 0.990 | 0.997 | 505.864 | 0.000 |
| Ocampo-Vázquez 2019 | 0.980 | 0.020 | 0.000 | 0.940 | 1.020 | 48.497 | 0.000 |
| Pissarra 2022 | 0.989 | 0.004 | 0.000 | 0.982 | 0.996 | 267.857 | 0.000 |
| | 0.975 | 0.006 | 0.000 | 0.962 | 0.987 | 156.778 | 0.000 |

Meta Analysis

**Fig 3. Prevalence of non-cardiac chest pain.** Forest plot illustrating the prevalence of non-cardiac chest pain across different studies, showing point estimates with 95% confidence intervals and associated statistical parameters. Test for overall effect; p-value > 0.001, Z-value = 156.7. Heterogeneity; p-value > 0.001, I2 = 98.7%.

## Prevalence of respiratory chest pain

| Study name | Point estimate | Standard error | Variance | Lower limit | Upper limit | Z-Value | p-Value | | Point estimate and 95% CI |
|---|---|---|---|---|---|---|---|---|---|
| Brancato 2021 | 0.192 | 0.040 | 0.002 | 0.114 | 0.270 | 4.850 | 0.000 | | |
| Drossner 2011 | 0.200 | 0.006 | 0.000 | 0.188 | 0.212 | 33.302 | 0.000 | | |
| Gesuete 2020 | 0.119 | 0.012 | 0.000 | 0.096 | 0.143 | 9.996 | 0.000 | | |
| Hambrook 2010 | 0.094 | 0.000 | 0.000 | 0.094 | 0.094 | 561.969 | 0.000 | | |
| Lin 2008 | 0.243 | 0.042 | 0.002 | 0.160 | 0.326 | 5.750 | 0.000 | | |
| Massin 2004 | 0.093 | 0.019 | 0.000 | 0.056 | 0.130 | 4.924 | 0.000 | | |
| Mohan 2018 | 0.035 | 0.004 | 0.000 | 0.026 | 0.044 | 7.811 | 0.000 | | |
| Pissarra 2022 | 0.128 | 0.012 | 0.000 | 0.105 | 0.151 | 10.823 | 0.000 | | |
| | 0.130 | 0.020 | 0.000 | 0.092 | 0.168 | 6.662 | 0.000 | | |

Favours A   Favours B

Meta Analysis

**Fig 4. Prevalence of respiratory chest pain.** Forest plot illustrating the prevalence of respiratory chest pain across different studies, showing point estimates with 95% confidence intervals and associated statistical parameters. Test for overall effect; p-value = 2.698, Z-value = 6.66. Heterogeneity; p-value> 0.001, $I^2$ = 98.6%.

## Musculoskeletal causes

Musculoskeletal chest pain was reported in nine studies with pooled prevalence = (0.357, 95% CI [0.202, 0.512]) [14–20, 22, 23]. I2 and p values were (99.8%, and p > 0.001, respectively) showing that pooled studies were heterogeneous. The forest plot for this outcome is shown in Fig 6.

## Psychogenic causes

The overall prevalence of psychogenic chest pain outcome pooled from six studies was (0.11, 95% CI [0.043, 0.177]) [14, 16, 17, 19, 22, 23]. The pooled studies were heterogeneous (I2 = 98%, p > 0.001). The forest plot for psychogenic chest pain outcome is illustrated in Fig 7.

## Idiopathic causes

Five studies reported idiopathic chest pain outcome among our studied population [14, 17, 18, 22, 23]. The pooled prevalence was (0.352, 95% CI [0.258, 0.446]). The pooled studies were

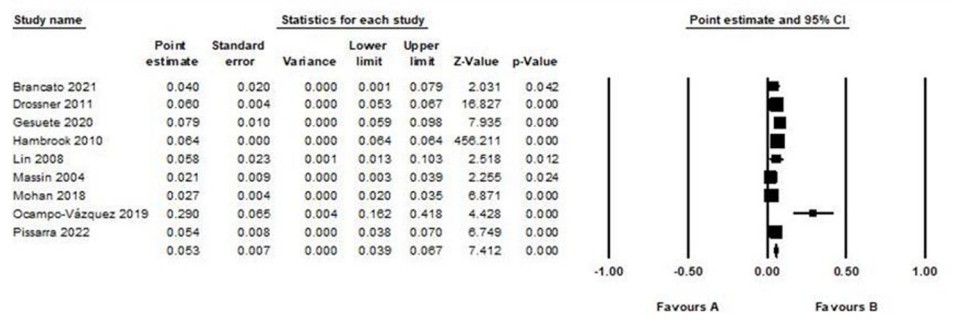

**Fig 5. Prevalence of gastrointestinal chest pain.** Forest plot illustrating the prevalence of gastrointestinal chest pain across different studies, showing point estimates with 95% confidence intervals and associated statistical parameters. Test for overall effect; p-value = 1.243, Z-value = 7.412. Heterogeneity; p-value> 0.001, $I^2$ = 98.4%.

## Prevalence of musculoskeletal chest pain

| Study name | Point estimate | Standard error | Variance | Lower limit | Upper limit | Z-Value | p-Value |
|---|---|---|---|---|---|---|---|
| Brancato 2021 | 0.202 | 0.040 | 0.002 | 0.123 | 0.281 | 5.006 | 0.000 |
| Drossner 2011 | 0.560 | 0.007 | 0.000 | 0.545 | 0.575 | 75.139 | 0.000 |
| Gesuete 2020 | 0.692 | 0.017 | 0.000 | 0.659 | 0.725 | 40.692 | 0.000 |
| Hambrook 2010 | 0.128 | 0.000 | 0.000 | 0.128 | 0.128 | 668.435 | 0.000 |
| Lin 2008 | 0.067 | 0.025 | 0.001 | 0.019 | 0.115 | 2.720 | 0.007 |
| Massin 2004 | 0.710 | 0.029 | 0.001 | 0.652 | 0.768 | 24.088 | 0.000 |
| Mohan 2018 | 0.113 | 0.008 | 0.000 | 0.098 | 0.128 | 14.635 | 0.000 |
| Ocampo-Vázquez 2019 | 0.415 | 0.071 | 0.005 | 0.276 | 0.554 | 5.835 | 0.000 |
| Pissarra 2022 | 0.330 | 0.017 | 0.000 | 0.297 | 0.363 | 19.825 | 0.000 |
|  | 0.357 | 0.079 | 0.006 | 0.202 | 0.512 | 4.520 | 0.000 |

**Fig 6. Prevalence of musculoskeletal chest pain.** Forest plot illustrating the prevalence of musculoskeletal chest pain across different studies, showing point estimates with 95% confidence intervals and associated statistical parameters. Test for overall effect; p-value = 6.198, Z-value = 4.52. Heterogeneity; p-value> 0.001, I2 = 99.8%.

heterogeneous with I2 and p-value = (96.6% and > 0.001, respectively). The forest plot for idiopathic chest pain outcome is illustrated in Fig 8.

## GRADE assessment

The certainty of evidence assessed in our systematic review is detailed in (S2 Table). According to GRADE, all outcomes were at a very low level of certainty. The causes of their downgrading were the heterogeneity of pooled studies in each assessed outcome and the publication bias as the observational studies are more attributable to it.

## Discussion

This systematic review showed important information regarding the etiology of chest pain in children. This is perhaps the first metanalysis that explores the causes of pediatric chest pain. The estimated incidence of sudden cardiac death in children ranges from 0.6 to 6.2 deaths per 100,000, with hypertrophic cardiomyopathy, coronary artery anomalies, and malignant

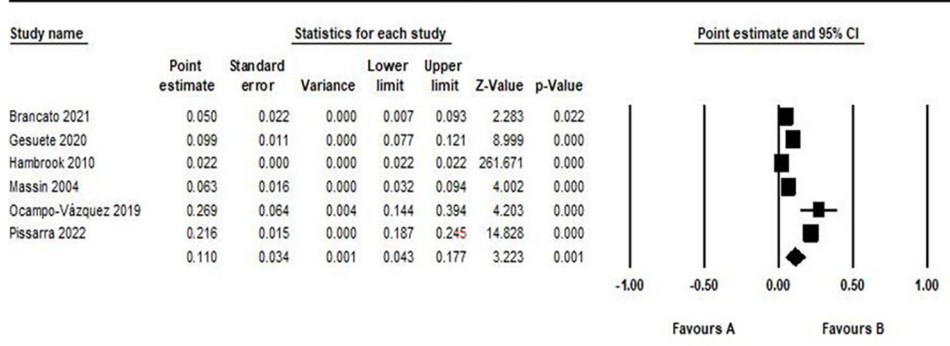

**Fig 7. Prevalence of psychogenic chest pain.** Forest plot illustrating the prevalence of psychogenic chest pain across different studies, showing point estimates with 95% confidence intervals and associated statistical parameters. Test for overall effect; p-value = 0.0012, Z-value = 3.22. Heterogeneity; p-value> 0.001, $I^2$ = 98%.

## Prevalence of idiopathic chest pain

| Study name | | | | | | | | Point estimate and 95% CI |
|---|---|---|---|---|---|---|---|---|
| | | | Statistics for each study | | | | | |
| | Point estimate | Standard error | Variance | Lower limit | Upper limit | Z-Value | p-Value | |
| Brancato 2021 | 0.454 | 0.050 | 0.003 | 0.356 | 0.552 | 9.073 | 0.000 | |
| Hambrook 2010 | 0.368 | 0.000 | 0.000 | 0.367 | 0.369 | 1331.307 | 0.000 | |
| Lin 2008 | 0.592 | 0.048 | 0.002 | 0.497 | 0.687 | 12.225 | 0.000 | |
| Ocampo-Vázquez 2019 | 0.124 | 0.048 | 0.002 | 0.031 | 0.217 | 2.607 | 0.009 | |
| Pissarra 2022 | 0.244 | 0.015 | 0.000 | 0.214 | 0.274 | 16.049 | 0.000 | |
| | 0.352 | 0.048 | 0.002 | 0.258 | 0.446 | 7.351 | 0.000 | |

Meta Analysis

**Fig 8. Prevalence of idiopathic chest pain.** Forest plot illustrating the prevalence of idiopathic chest pain across different studies, showing point estimates with 95% confidence intervals and associated statistical parameters. Test for overall effect; p-value = 1.97, Z-value > 7.35. Heterogeneity; p-value> 0.001, $I^2$ = 96.6%.

arrhythmias being responsible for many cases, while coinciding with athletic activities [24]. The risk and fear of misdiagnosing can create stressful scenarios for healthcare providers. This can lead to excessive diagnostic testing, which can create poor patient experiences as well as place a great financial burden on the healthcare system. Additionally, determining which children are at risk for sudden cardiac death is challenging due to the frequent lack of alarming symptoms.

Chest pain is common in children and accounts for 0.6% of pediatric emergency department visits [25]. Chest pain always raises concern of cardiac abnormalities, but it is rarely cardiac in origin. When it is, the consequences of misdiagnosis are quite serious. Accurate diagnosis is required to prevent unnecessary emergency department evaluation and cardiology referral. In most cases, after obtaining a thorough history and focused physical examination, no immediate intervention is needed. However, some children may have serious and life-threatening conditions, and a careful evaluation is required in every child presenting with chest pain to determine whether a cardiac condition may exist and guide the evaluation strategy [26]. In general, chest pain associated with exercise, syncope, or palpitations requires further evaluation and workup.

This study showed that cardiac causes of chest pain are rare in children. More specifically, our study shows that only 2.5% of our study population had chest pain that was secondary to a cardiac cause. This supports the previous findings that cardiac-related causes for chest pain in children are very rare [7, 25, 27]. It is worth noting that our review found variation in the specific cardiac pathologies among the included studies. This could have been due to several different factors such as patient location, demographics, institution of care, etc. Pericardial disease was found to be a cause in all three of the studies that mentioned pathological cardiac etiologies, however it was not the most prevalent in all three studies[27, 28].

Regarding specific pediatric cardiac causes, arrhythmias are implicated in many cardiac-related presentations [29]. Supraventricular tachycardia is the most common rhythm disturbance, and ventricular tachycardia or bradycardias are the most ominous rhythms [29]. These presentations are serious and require urgent evaluation and pediatric cardiology consultation.

Our review found 97.5% of our study population have a non-cardiac induced presentation of chest pain. The alternative causes of chest pain reported were respiratory, gastrointestinal, psychogenic, musculoskeletal, and idiopathic. The most prevalent non-cardiac cause of chest pain was musculoskeletal followed by idiopathic and then gastrointestinal. This is consistent with previous studies which have found these causes to be among the most common non-cardiac causes of pediatric chest pain [7, 8, 27]. Despite that there was still variation in the

prevalence of these non-cardiac causes in the included studies. The cause of chest pain in many children remains unknown and warrants further investigation for a clearer understanding of pediatric chest pain etiology.

Our review also analyzed reports of chest pain location and radiation of pain to determine common presentations of children with cardiac pathologies. Among the studies that reported this information, the most common presentation of chest pain was retrosternal chest pain. Diffuse chest pain was the second most common reported type of pain. To our knowledge this is the first review to report chest pain location and radiation amongst a study population.

Our findings have clinical relevance and implications as algorithm-based policies are expanding nationally to limit reimbursement for visits for non-emergent causes. These algorithms derive emergent need for ED visits from billing codes rather than the presenting symptoms. It is therefore important to limit the diagnostic workup and reserve it for patients with positive findings on initial assessment. A set of red-flag criteria have been determined to help in the assessment of patients with chest pain [30]. Some red-flag criteria include, but are not limited to, exertional chest pain, exertional syncope, chest pain radiation, and past medical history of sudden cardiac death and hypercoagulable state. If a patient does not exhibit any red-flag criteria at all, then it is likely that the etiology of their chest pain is non-cardiac.

## Limitations

This systematic review has several limitations. First, there was significant heterogeneity among the included studies. Second, many studies were retrospective, which did not allow the authors to determine the precise cause of chest pain due to the clinicians not being present at the time of presentation. These studies also have a small patient population with limited follow-up. Furthermore, these results should be interpreted cautiously because only eleven studies could be included in this review. This may be related to our reliance on a relatively limited number of databases for the identification of potentially eligible research studies. Also, it is susceptible to selection bias. There is always a potential for misclassification. For example, a patient may have a significant diagnosis but may be coded with that specific diagnosis rather than the more general chest pain.

## Future considerations

In our systematic review, it was noted that the prevalence of cardiac pathologies differed among the included studies. The studies that reported on this all differed in location, demographics, and institution of care. A similar observation was made for the prevalence of non-cardiac causes of chest pain as well. These observations warrant more research to further understand the causes of such variations among study populations.

None of the studies that we reviewed categorized the chest pain location by the specific cause of chest pain. Further investigation is needed in this area of study as it could prove to be an essential tool in the practice of ED physicians who encounter pediatric patients with a chief complaint of chest pain. We also noticed heterogeneity of studies.

## Conclusions

Chest pain in children, an infrequent occurrence, typically presents as benign in the realm of pediatric emergency departments and cardiologist's offices. Within this context, the approach to evaluation is marked by its simplicity and efficacy. A comprehensive history-taking, meticulous physical examination, and a judicious screening electrocardiogram (ECG) often prove to be the sole requisites for excluding the exceedingly rare, yet potentially life-threatening, causes of chest pain in the pediatric population.

The use of laboratory tests remains a selective strategy, triggered only when warranted by the clinical presentation and the findings of the physical assessment. This approach prioritizes precision and resource optimization in the diagnostic process.

Through this study, we aim to enrich available resources at the disposal of physicians, empowering them to navigate the intricacies of assessing pediatric patients with chest pain presentations. By embracing this innovative approach, we hope to enhance the quality of care delivered while concurrently curbing the utilization of unnecessary investigations, thereby advancing pediatric medicine.

## Practice points

Most children present with chest pain do not have an underlying cardiac cause. Although evaluation of chest pain is extensive, it rarely yields cardiac etiology. Extensive workup may not be needed in patients for whom a clear etiology, other than cardiac disease, can be determined.

## Declarations

We confirm that the manuscript has been read and approved by all named authors and that there are no other persons who satisfy the criteria for authorship but are not listed.

We further confirm that the order of authors listed in the manuscript has been approved by all of us.

We confirm that we have given due consideration to the protection of intellectual property associated with this work and that there are no impediments to publication, including the timing of publication, with respect to intellectual property. In doing so, we confirm that we have followed the regulations of our institutions concerning intellectual property.

## Supporting information

**S1 Checklist. PRISMA checklist.**
(DOCX)

**S1 Appendix. Literature search.**
(DOCX)

**S1 Table. Assessing the quality of studies and evaluating the risk of bias using the Newcastle-Ottawa Quality Assessment Scale.**
(PDF)

**S2 Table. Assessment of certainty for the etiologies of chest pain.** This table shows the certainty for the etiologies of chest pain.
(PDF)

## Author Contributions

**Conceptualization:** Mohammed Alsabri, Alaa Ahmed Elshanbary, Mohamed Sayed Zaazouee.

**Formal analysis:** Anas Zakarya Nourelden, Ahmed Hashem Fathallah.

**Investigation:** Jorge Pincay, Zaid Nakadar, Lita Aeder.

**Methodology:** Jorge Pincay, Zaid Nakadar.

**Supervision:** Mohammed Alsabri, Mohamed Sayed Zaazouee, Lita Aeder.

**Writing – original draft:** Mohammed Alsabri, Alaa Ahmed Elshanbary, Anas Zakarya Nour-elden, Ahmed Hashem Fathallah, Mohamed Sayed Zaazouee, Muhammad Wasem.

**Writing – review & editing:** Mohammed Alsabri, Anas Zakarya Nourelden, Ahmed Hashem Fathallah, Mohamed Sayed Zaazouee, Muhammad Wasem, Lita Aeder.

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
