## [Decision Letter · Decision Letter 0]

25 Sep 2023

PONE-D-23-22358Presentation, Risk Factors, and Outcomes of Pediatric Patients who Present to the Emergency Department with Chest Pain: a systematic review and meta-analysisPLOS ONE

Dear Dr. Alsabri,

Thank you for submitting your manuscript to PLOS ONE. After careful consideration, we feel that it has merit but does not fully meet PLOS ONE’s publication criteria as it currently stands. Therefore, we invite you to submit a revised version of the manuscript that addresses the points raised during the review process.

We look forward to receiving your revised manuscript.

Kind regards,

Antoine Fakhry AbdelMassih

Academic Editor

PLOS ONE

Journal Requirements:

2. Please include a separate caption for each figure in your manuscript.

3. Please include your tables as part of your main manuscript and remove the individual files. Please note that supplementary tables (should remain/ be uploaded) as separate ""supporting information files"""

4. We notice that your supplementary tables are included in the manuscript file. Please remove them and upload them with the file type 'Supporting Information'. Please ensure that each Supporting Information file has a legend listed in the manuscript after the references list.

5. Please include a copy of Table 3 which you refer to in your text on page 8.

Reviewers' comments:

Reviewer's Responses to Questions

**Comments to the Author**

1. Is the manuscript technically sound, and do the data support the conclusions?

Reviewer #1: Yes

2. Has the statistical analysis been performed appropriately and rigorously? 

Reviewer #1: Yes

3. Have the authors made all data underlying the findings in their manuscript fully available?

Reviewer #1: Yes

4. Is the manuscript presented in an intelligible fashion and written in standard English?

Reviewer #1: No

5. Review Comments to the Author

Reviewer #1: 1. The “Title” is inaccurate and incorrect. The “Title” modification is needed. Please, it is preferable to change it to “Chest Pain in Pediatric Patients in the Emergency Department-Presentation, Risk Factors and Outcomes-A Systematic Review and Meta-analysis”.

2. Regarding the initial phrase of the “Objective” of the “Abstract”; (This study sought to assess and determine…) is inappropriate. Please, change it to ‘This study aimed to assess and determine…).

3. The author (s) mentioned in the “Conclusion” of the “Abstract” this sentence (This review found musculoskeletal pain to the be most common cause of chest pain in children.) then the author (s) in mentioned this sentence (Review of our study population identified retrosternal chest pain to be the most common location of pain.). Please, both musculoskeletal pain and retrosternal chest pain are relevant to the entity of “location of pain” no more. So, both of the above sentences are contradictory and scientifically unacceptable. Please, review both and edit them.

4. The “Conclusion” of the “Abstract” is scientifically very weak. Re-writing it again is needed.

5. In the “Introduction”, for this sentence, “Statistically, many children with chest pain have gastrointestinal etiologies with referred chest pain”, please, support your sentence with numbers and percentages.

6. In the “Introduction”, also for this sentence, “Statistically, many children with chest pain have gastrointestinal etiologies with referred chest pain” is proceeded and followed with general writing sentences on “chest pain”. This is not a good understanding flow. The author (s) should mention some sentences regarding general “chest pain or chest pain in adults” and then start to specify writing on the “chest pain in pediatrics” or use a comparative method between “chest pain in children and adults”.

7. Naming of investigators; even in abbreviations and their work in this manuscript (M.A., Z.N., J.P., M.S.Z created the review protocol and a search strategy) in the “Methods” is a form of misplacement. So, please, transfer this sentence “Investigators (M.A., Z.N., J.P., M.S.Z) created the review protocol and a search strategy.” From “Methods” to the “Author contribution statement” section and change accordingly. Also, transfer this sentence “Two assessors (Z.N. and J.P.) independently rated the quality of the studies using the Cochrane Collaboration’s tool”. From “Quality assessment of studies” to the “Author contribution statement” section and change accordingly.

8. Please, where is the reference for (Figure 1) in the text?!

9. The following sentence in the “Limitations”; (Second, many studies were observational and retrospective, which did not allow the authors to determine the precise cause of chest pain) is incorrect and needs literature reviewing.

10. Regarding the final “Conclusion” it must be decisive, innovative, and expressive for the article. The “Conclusion” of the “Abstract” is also a brief reflective mirror of it.

11. Otherwise the included 2 studies (Drossner 2011; reference No. 6) and Massin 2004; reference No. 9). Where are the final references and their citation in the tables for the remaining included studies?!

12. There are several “Grammatical” mistakes. So, corrections and language editing are essential.

6. PLOS authors have the option to publish the peer review history of their article (what does this mean?). If published, this will include your full peer review and any attached files.

Reviewer #1: No

---

## [Author Response · Author response to Decision Letter 0]

11 Oct 2023

Response to Journal Requirements:

1. Thank you very much for your kindness in referencing the PLOS ONE style templates. We have reviewed them and adjusted the styling and format of the manuscript accordingly! Please let us know if there are still segments that are still not meeting the journal standards of formatting and style. Thank you!

2. Each figure was reviewed and ensured that a separate caption was given to each one.

3. We have pasted the tables as part of the main manuscript and removed the individual files. The supplementary tables have remained their own separate files. 

4. The supplementary files were made into separate files and their legend was included at the end of the manuscript as per suggestion. Thank you!

5. This issue was resolved by removing the statement that referenced Table 3 as this was a mistake. We apologize for the confusion!

6. The caption and legends for the supplemental information was added to the end of the manuscript. Additionally we have updated some of the references accordingly with the journal style of formatting. Thank you for your guidance in this issue!

7. No additions or removals were made to the citation list. 

Thank you so very much for the kind suggestions for the manuscript.

Response to reviewers

1. The title was changed as per the suggestion of the reviewer. Thank you!

2. This sentence was changed as per suggestions of the reviewer. Thank you!

3. We are having a little bit of difficulty understanding this point that was brought up. We understand musculoskeletal pain to be an etiology of pain as opposed to a means of localizing the pain. Retrosternal pain indicates a location in which any type of pain may be present. We are open to discussion and would love to further understand this issue to clarify it as much as possible for readers. Thank you for your help!

4. The conclusion of the abstract was re-written to be stronger. Thank you for pointing this out!

5. A reference was added to support this point. Thank you!

6. This portion of the introduction was changed as per suggestions of the reviewer. Thank you for advising us in making this section more coherent and better in flow!

7. The mentioned sentences were removed from their original locations and added to the author contributions portion of the manuscript. Thank you for clarifying this for us and making this wonderful suggestion!

8. The reference for Figure 1 was added into the text. Thank you!

9. This sentence was edited and the word observational was removed as there was some misunderstanding in this part. We meant to say only retrospective as this would make more sense because any physicians conducting the study would not be present at the time of patient encounter to directly examine the patients. 

10. The conclusion was re-written to fall more in line with the reviewers suggestions. Thank you very much!

11. We have added in the references that were referred to in tables 1 and 2 that were used to conduct this study. Our apologies for this very obvious blunder. Thank you for your help in correcting this. 

12. The manuscript was edited to correct for any grammatical errors.

---

## [Decision Letter · Decision Letter 1]

2 Nov 2023

Presentation, Risk Factors, and Outcomes of Pediatric Patients who Present to the Emergency Department with Chest Pain: a systematic review and meta-analysis

PONE-D-23-22358R1

Dear Dr. Alsabri,

We’re pleased to inform you that your manuscript has been judged scientifically suitable for publication and will be formally accepted for publication once it meets all outstanding technical requirements.

Kind regards,

Antoine Fakhry AbdelMassih

Academic Editor

PLOS ONE

Additional Editor Comments (optional):

The final comments of reviewer 1, are mainly focused on reference formatting, which can be handled during proofing. 

Thank you for fulfilling most of the required edits.

Reviewers' comments:

Reviewer's Responses to Questions

**Comments to the Author**

1. If the authors have adequately addressed your comments raised in a previous round of review and you feel that this manuscript is now acceptable for publication, you may indicate that here to bypass the “Comments to the Author” section, enter your conflict of interest statement in the “Confidential to Editor” section, and submit your "Accept" recommendation.

Reviewer #1: All comments have been addressed

2. Is the manuscript technically sound, and do the data support the conclusions?

Reviewer #1: Yes

3. Has the statistical analysis been performed appropriately and rigorously? 

Reviewer #1: Yes

4. Have the authors made all data underlying the findings in their manuscript fully available?

Reviewer #1: Yes

5. Is the manuscript presented in an intelligible fashion and written in standard English?

Reviewer #1: Yes

6. Review Comments to the Author

Reviewer #1: For the Author.

Thank you to the authors for the positive response in the correction of many of the reviewer comments. But, there are still few comments.

1. Unfortunately, the “Title; lines 1-3” is still unchanged. I don’t know why?! The “Title and abstract” are the face of the article, interestingly, attractive for readers, Journals, Conferences selection for presentation, and also a mirror for the authors.

2. Regarding the following sentence in the “Background”, “Statistically, many children with chest pain have gastrointestinal etiologies with referred chest pain”.

Regrettably, the author changed it to “However, the landscape shifts when considering pediatric cases, where statistical evidence points to gastrointestinal origins often leading to referred chest pain; lines 58-60” with no response to the reviewer's request (please, support your sentence with numbers and percentages). I don’t know, why? And there is no more difference between the first and the modified sentence.

3. The author (s) added (reference No. 14; line 60) to the above sentence “However, the landscape shifts when considering pediatric cases, where statistical evidence points to gastrointestinal origins often leading to referred chest pain [14]” as the first reference in the text citation How?! This is unacceptable. Please, arrange the references in serial order (1, 2, 3, 4,5…etc.). Serial order of the reference citing in the text is essential.

4. Regarding the addition of the defective references that were referred to in (Tables 1 and 2), thank you. But why do you neglect to add each No. of reference in brackets [] opposite each study ID in the tables rows themselves. Please, add the No. of reference in brackets [] in the right place of the tables.

5. Regarding the “Quality assessment of studies”, the reference citing (Kirmayr et al., 2021; lines 150) is a missed in the final references. Please, add it to the final reference list and put it in the text as a number in brackets [] with start this sentence with Kirmayr et al.

7. PLOS authors have the option to publish the peer review history of their article (what does this mean?). If published, this will include your full peer review and any attached files.

Reviewer #1: **Yes: **Dr. Yasser Mohammed Hassanain Elsayed

---

## [Editor Report · Acceptance letter]

26 Feb 2024

PONE-D-23-22358R1 

PLOS ONE

Dear Dr. Alsabri, 

I'm pleased to inform you that your manuscript has been deemed suitable for publication in PLOS ONE. Congratulations! Your manuscript is now being handed over to our production team.

Kind regards, 

on behalf of

Prof Antoine Fakhry AbdelMassih 

Academic Editor

PLOS ONE